# Atomic-scale manipulation of polar domain boundaries in monolayer ferroelectric In$_2$Se$_3$

Fan Zhang [1,7,10], Zhe Wang[2,3,8,10], Lixuan Liu[4,5], Anmin Nie [5], Yanxing Li[6], Yongji Gong [4], Wenguang Zhu [2,3] ✉ & Chenggang Tao [1,9] ✉

Domain boundaries have been intensively investigated in bulk ferroelectric materials and two-dimensional materials. Many methods such as electrical, mechanical and optical approaches have been utilized to probe and manipulate domain boundaries. So far most research focuses on the initial and final states of domain boundaries before and after manipulation, while the microscopic understanding of the evolution of domain boundaries remains elusive. In this paper, we report controllable manipulation of the domain boundaries in two-dimensional ferroelectric In$_2$Se$_3$ with atomic precision using scanning tunneling microscopy. We show that the movements of the domain boundaries can be driven by the electric field from a scanning tunneling microscope tip and proceed by the collective shifting of atoms at the domain boundaries. Our density functional theory calculations reveal the energy path and evolution of the domain boundary movement. The results provide deep insight into domain boundaries in two-dimensional ferroelectric materials and will inspire inventive applications of these materials.

Domain boundaries refer to interfaces within a material between adjacent domains with distinct crystallographic orientations. Owing to symmetry breaking or topological protection, these boundaries can exhibit substantially different properties compared to their parent materials. One specific category is polar domain boundaries in ferroelectric materials, separating uniformly polarized domains. Despite polar domain boundaries hold significant potential for broad applications in nanoelectronics and quantum information technology and have been intensively explored[1–11], microscopic understanding and tuning of polar domain boundaries are still lacking[1], especially on how to manipulate the domain boundaries at the atomic scale. Among the manipulation methods[12–14], applying an external electric field is a common way to manipulate ferroelectric boundaries, which can also

lead to numerous potential applications of domain boundaries in future quantum devices. At the nanoscale, this method can be achieved through using a piezoresponse force microscope[15–18], transmission electron microscope[19–21], or scanning tunneling microscope (STM)[22]. As a powerful characterization tool, STM is capable of not only probing both atomic structures and local electronic properties but also manipulating atoms and atomic scale features. However, due to the challenges associated with preparing cross-section samples and eliminating impurities that are energetically favorable to accumulate at polar domain boundaries, conducting STM studies on domain boundaries in bulk ferroelectric materials like BiFeO$_3$ has proven highly difficult[23,24]. With their lower dimensionality, two-dimensional (2D) ferroelectric materials offer a new and ideal platform for

[1]Department of Physics, Virginia Tech, Blacksburg, VA 24061, USA. [2]International Center for Quantum Design of Functional Materials (ICQD), Hefei National Research Center for Physical Sciences at the Microscale, University of Science and Technology of China, Hefei 230026, China. [3]Department of Physics, University of Science and Technology of China, Hefei 230026, China. [4]School of Materials Science and Engineering, Beihang University, Beijing 100191, China. [5]Center for High Pressure Science, State Key Laboratory of Metastable Materials Science and Technology, Yanshan University, Qinhuangdao 066004, China. [6]Department of Physics, University of Texas at Austin, Austin, TX 78712, USA. [7]Present address: Department of Physics, University of Texas at Austin, Austin, TX 78712, USA. [8]Present address: Department of Physics, Southern University of Science and Technology, Shenzhen 518055, China. [9]Present address: Center for Nanophase Materials Sciences, Oak Ridge National Laboratory, Oak Ridge, TN 37830, USA. [10]These authors contributed equally: Fan Zhang, Zhe Wang. ✉e-mail: wgzhu@ustc.edu.cn; cgtao@vt.edu

investigating and tuning polar domain boundaries, overcoming the limitations faced in bulk materials. Several recent studies demonstrated the possibility of tip bias induced domain or polarization switching in 2D ferroelectric materials[22,25–27]. Here we used STM to manipulate the domain boundaries in 2D In$_2$Se$_3$, an emerging 2D ferroelectric material[28–35]. Atomically thin In$_2$Se$_3$ was synthesized through the chemical vapor deposition (CVD) method. We manipulate various types of domain boundaries in monolayer $\beta'$ In$_2$Se$_3$ with atomic precision and study the detailed domain boundary dynamics during manipulation by using STM. The results show that the domain boundaries move in a step-by-step manner. The underlying mechanism is explained by the energy pathway calculated through the DFT calculations. Our findings, which elucidate the internal structure and unveil the dynamic behaviors of polar domain boundaries in two-dimensional ferroelectric materials, provide crucial insight for fundamental research on ferroelectric domain boundaries and open avenues for applications in data storage and electronic devices that leverage nanoscale ferroelectricity and localized functional properties.

## Results

### Polar domain boundaries in monolayer $\beta'$ In$_2$Se$_3$

The atomically thin In$_2$Se$_3$ films were synthesized on highly oriented pyrolytic graphite (HOPG) using the chemical vapor deposition (CVD) method[29,36], and all scanning tunneling microscopy (STM) measurements were conducted at 77 K. It has been previously reported that at this temperature, In$_2$Se$_3$ stabilizes in the $\beta'$ phase[29]. Figure 1a illustrates the atomic model of the $\beta'$ In$_2$Se$_3$ structure, while Fig. 1b presents an atomically-resolved STM image of monolayer $\beta'$ In$_2$Se$_3$, with the unit cell outlined by a black rectangle. Surface Se atoms within a unit cell are depicted on the STM image, using shades of color to indicate the relative heights of Se atoms, where dark red signifies higher, while light red signifies lower. The side view of the atomic model in Fig. 1a, highlighted by the dashed rectangle, reveals that surface Se atoms within a unit cell are at different heights, with the highest ones positioned at the corners of the unit cell. The two central surface Se atoms inside a unit cell are arranged in a manner where one is higher than the other.

Density functional theory (DFT) calculations indicate an in-plane polarization with a magnitude of 1.517 eÅ per In$_2$Se$_3$ ($1.7 \times 10^{-10}$ C m$^{-1}$) along the $a$ axis. This polarization is pointing from the higher central atom to the lower one, as denoted by the green arrows. The in-plane ferroelectricity of the $\beta'$ In$_2$Se$_3$ stems from the displacement of the central-layer Se atoms from their highly symmetric positions, as illustrated in Fig. 1a. This figure elucidates the nature of the in-plane electric polarization in $\beta'$ In$_2$Se$_3$. Due to the reduced symmetry of the $\beta'$ phase that originates from the $\beta$ phase[29], various polar domain boundaries have been observed between domains with different relative rotational angles in the $\beta'$ phase[36]. The majority of the observed domain boundaries are 60° and 120°, with a distribution of ~72% for the 60° domain boundaries and ~27% for the 120° ones. Rarely, other types of domain boundaries that can not be well defined or can be interpreted as structural line defects that separate domains (one of them shown in Supplementary Fig. 1) were observed, which account for ~1% of all the observed domain boundaries. In the following sessions, we focus on the 60° domain boundaries since they are abundant and usually straighter and more ordered than the 120° ones[36]. Based on

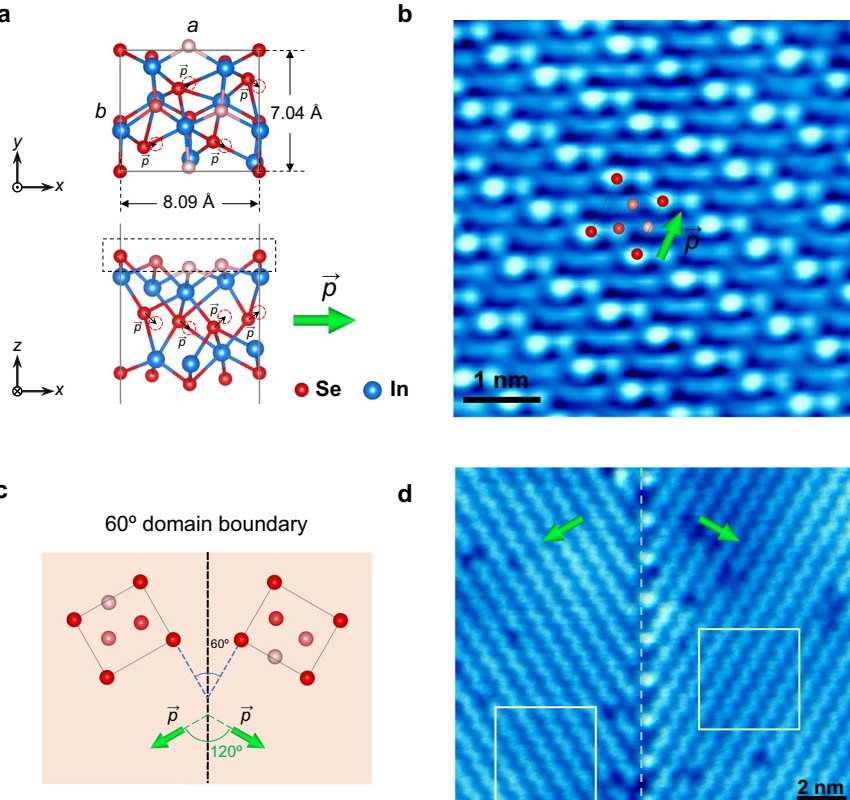

**Fig. 1 | Domain boundaries in 2D $\beta'$ In$_2$Se$_3$. a** Atomic model of $\beta'$ In$_2$Se$_3$ structure. Red dashed circles indicate the symmetry position of central layer Se atoms. Black arrows denote dipoles induced by the central layer Se atoms deviating from the symmetry position. The green arrow indicates the directions of the in-plane electric polarization. **b** An atomically-resolved STM image of monolayer $\beta'$ In$_2$Se$_3$ ($V_S = 1.0$ V, $I = 0.4$ nA). A unit cell is marked and overlapped with the surface Se atoms from the atomic model, highlighted by the dashed rectangle in the side view in (**a**). The shades of the color indicate the relative height of each Se atom. **c** Illustration of the 60° tail-to-tail domain boundary. The blue dashed lines are along the b directions of the unit cell in each domain, forming a 60° angle. The boundary serving as the angle bisector is marked with black dashed lines. The angle between the polarization vectors of adjacent domains is indicated in green. **d** Large scale STM image showing a 60° domain boundary in monolayer In$_2$Se$_3$ ($V_S = 4.2$ V, $I = 0.2$ nA).

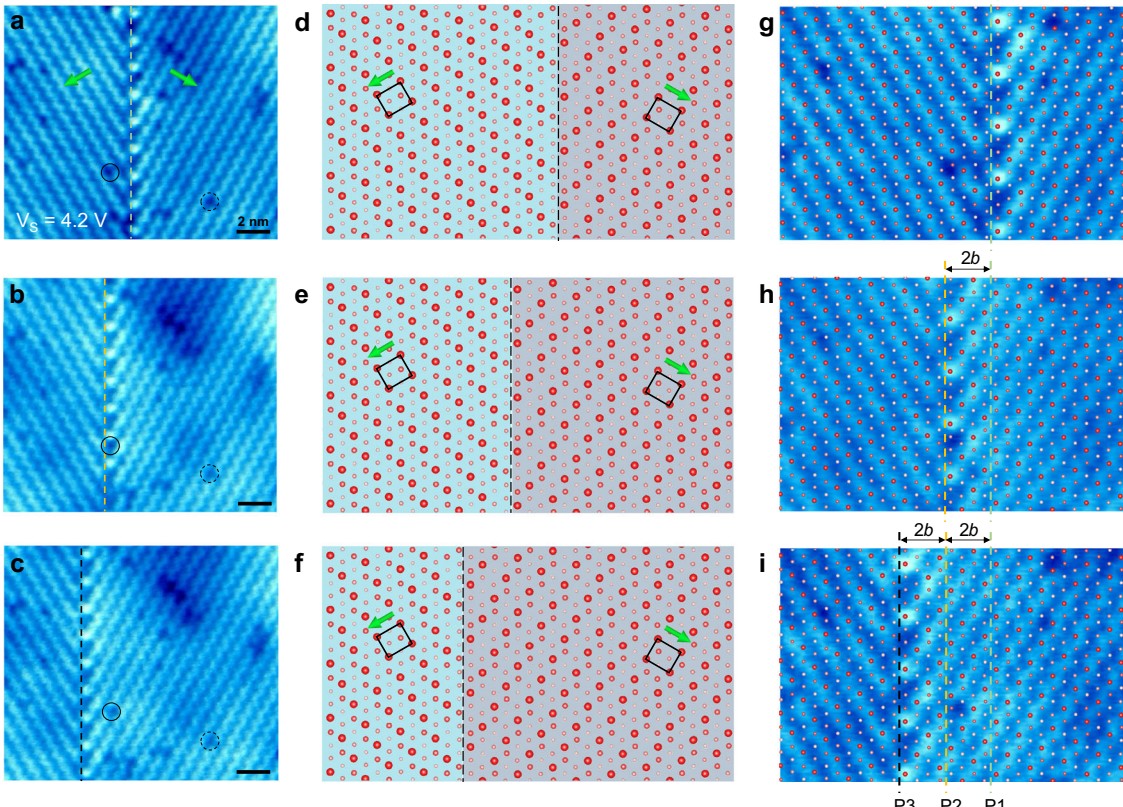

**Fig. 2 | Manipulation of a tail-to-tail domain boundary in monolayer $\beta'$ In$_2$Se$_3$.**
**a** STM image of a 60° tail-to-tail domain boundary in monolayer $\beta'$ In$_2$Se$_3$. The green arrows indicate the polarization direction of each domain. The initial position of the domain boundary is marked by the green dashed line. **b**, **c** The following two positions of the domain boundary. The defect marked by the dashed circle in **a**–**c** serves as a reference point. The domain boundary positions are indicated by yellow and black dashed lines in (**b**) and (**c**), respectively. Scanning parameters for **a**–**c** $V_S$ = 4.2 V, $I$ = 0.2 nA, and Scanning speed = 600 s per image. The scale bars are 2 nm. **d**–**f** Calculated atomic models of a 60° tail-to-tail domain boundary with marked unit cells and polarization. **g**–**i** zoom-in images of **a**–**c** overlapped with the atomic models.

---

the polarization directions of neighboring domains, the domain boundaries can be divided into the head-to-tail, tail-to-tail, and head-to-head types. Figure 1c shows the domain configuration for the 60° tail-to-tail boundaries. The angle between the polarization vectors of adjacent domains is indicated in green. The boundary serving as the angle bisector is marked with black dashed lines. Figure 1d is a large-scale STM image with a 60° domain boundary. The tail-to-tail type is determined by the zoom-in STM images obtained in the green boxes on each domain in Fig. 1d (see Supplementary Fig. 2a, b in the Supporting Materials).

**Manipulation of domain boundaries with atomic precision**

In addition to visualizing the domain boundaries, we further use STM to manipulate the boundaries in a controllable way. Figure 2a shows a 60° tail-to-tail domain boundary (the same boundary shown in Fig. 1d) with the initial position marked by the dashed green line along the brightest Se atoms on the domain boundary. Under a sample bias of 4.2 V and continuous scanning, the electric field from the STM tip is able to drive the domain boundary to move, as shown in STM images in Fig. 2a, b. Here the defects in the vicinity, two of which are highlighted by the dashed and solid circles in Fig. 2a, serve as the reference points for measuring the movement of the domain boundary. The sequential STM images reveal that the 60° domain boundary moves in a step-by-step manner from the initial position to the new positions marked by a dashed yellow line in Fig. 2b and a dashed black line in Fig. 2c, respectively. Although the domain boundary prefers a straight configuration, during the moving process, parts of the domain boundary usually move first, and then the

whole boundary reaches the new position (Supplementary Fig. 3 and Supplementary Movie 1). In most cases, the section moves from right to left, as shown in Fig. 2a–c. The opposite motion, from left to right, is occasionally observed during the process. That implies a slight difference between the energy barriers for the transition of the domain boundary to the left and the right. Figure 2d–f are the atomic models of a 60° tail-to-tail domain boundary, with the center of the domain boundary marked by the dashed line. To clearly show the domain boundary movement in Fig. 2d–f, the left and right domains are respectively colored in light blue and light purple. The zoom-in images of Fig. 2a–c, overlapped with the atomic models, are shown in Fig. 2g–i, in which the domain boundary positions are labeled as P1, P2, or P3. The domain boundary moves from one position to the nearest neighboring position while keeping the same configuration of the domain boundary. The step length between the nearest neighboring positions marked in Fig. 2g–i is 2b, where b is the lattice constant along the b axis. This specific step length is consistent with a translational symmetry of the domain boundary structure (see Supplementary Fig. 4). The translational vector is indicated by the long black arrow across the domain boundary pointing from the initial atom position to the final position during one-step movement. This explanation is supported by the observation of the movement of parts of the boundary (Fig. 2 and Supplementary Fig. 3). It is worth noticing that the movement of the domain boundary is not pinned by some types of point defects, one of which is highlighted by the black solid circles in Fig. 2a–c. The overall biased motion, in this case from right to left, should be related to the asymmetry in structural and electronic structures of the domain boundary along the direction

perpendicular to the domain boundary, as described in our previous work[36].

To explore how the polarization configurations affect the manipulation of domain boundaries, we now move to 60° head-to-tail domain boundaries. Figure 3a shows a 60° head-to-tail domain boundary with the initial position marked by the dashed yellow line along the atoms that appear brightest at the domain boundary. The insets show the zoom-in images of the two neighboring domains by which the polarization of each domain is determined. Under a sample bias of 3.2 V and continuous scanning, the electric field from the STM tip is able to drive this domain boundary to move, as shown in a series of STM images in Fig. 3a–f. Overall, similar to 60° tail-to-tail domain boundaries, the 60° head-to-tail domain boundary moves in a step-by-step manner from the initial position marked by the dashed yellow line in Fig. 3a to a new position marked by the dashed black line in Fig. 3f. Figure 3g, h are the atomic models of a 60° head-to-tail domain boundary with marked unit cells and polarization. The dashed line marks the center of the domain boundary. To clearly show the domain boundary movement from Fig. 3g–h, the left and right domains are highlighted with light blue and light purple backgrounds, respectively. Same as the 60° tail-to-tail domain boundaries, the step length between the nearest neighboring positions is 2*b*. That also attributes to the translational symmetry of the head-to-tail domain boundary structure.

Although the 60° head-to-tail domain boundaries exhibit similar movement behaviors to the 60° tail-to-tail domain boundaries, the threshold of the bias that triggers the domain boundaries' motion is much smaller than the 60° tail-to-tail domain boundaries. With similar domain boundary moving dynamics, the threshold bias for 60° head-to-head domain boundaries (Supplementary Fig. 5) is close to that for tail-to-tail type. In our experiments, we observed that the averaged threshold bias for the 60° head-to-tail domain boundaries is around 2.1 V, while for the 60° tail-to-tail (or head-to-head) domain boundaries, the averaged threshold bias is around 3.4 V (see more detailed statistic in Supplementary Fig. 6).

### Difference of manipulating boundaries with distinct polar configurations

To understand the difference in the threshold bias of manipulating different domain boundaries, we simulated the movement of the tail-to-tail and head-to-tail domain boundaries with simplified models, respectively. As shown in Fig. 2a, b, during the movement of the tail-to-tail domain boundary, only the polarization direction of the left region adjacent to the domain boundary changes (the polarization direction changes from down left to down right). Therefore, it is reasonable to simulate the movement of the tail-to-tail domain boundary by the polarization reversal. As shown in Fig. 4a, the initial and final states, respectively, correspond to the atomic structures of the left and right domains in Fig. 2a. The polarization directions of the initial and final states are opposite in the *x*-axis and the same in the *y*-axis, both along the negative direction of the *y*-axis. It is conceivable that in the process of polarization reversal, the magnitude of the polarization first decreases and then increases in the *x*-axis, while the polarization persists in the *y*-axis. The kinetic calculations of the polarization reversal also support our speculations. In the transition state of Fig. 4a, the central-layer Se atoms are uniformly deviated from the symmetry position (the center of the hexagon) along the positive direction of the *y*-axis, implying an in-plane electric polarization along the negative direction of the *y*-axis.

A similar analysis can be done for the head-to-tail domain boundary. As shown in Fig. 3, the movement of the boundary results in the polarization direction changing from up left to down left in the left region near the domain boundary. Similarly, we simulated the movement of the head-to-tail domain boundary by this polarization inversion process, as shown in Fig. 4b. The polarization directions of the initial and final states are opposite in the *y*-axis and the same in the *x*-axis, along the negative direction of the *x*-axis. For the transition state, the central-layer Se atoms are uniformly deviated from the symmetry position (the center of the hexagon) along the positive direction of the *x*-axis, indicating an in-plane electric polarization along the negative direction of the *x*-axis.

However, the calculated energy barriers are similar (~0.03 eV per In$_2$Se$_3$) for the movement of the tail-to-tail or the head-to-tail domain boundary. In this work, the movement of domain boundaries in the experiment is driven by an external electric field induced by the STM tip (see the detailed simulation of the electric field in Supplementary Fig. 7). Therefore, it is necessary to explore the effect of the external electric field on the barrier. The calculation results under the various external electric field are shown in Fig. 4c, where the energy barrier was calculated by the energy difference between the transition state and the initial state. It is clear that the energy barrier of the domain boundary movement at the tail-to-tail is insensitive to the external electric field, while it decreases rapidly as the electric field increases at the head-to-tail. This result is consistent with the experimental phenomenon that the head-to-tail domain boundary is easier driven by the external electric field, which also confirms the validity of our previous analysis that approximates the energy barrier of domain boundary shift in the large supercell by in-plane polarization reversal in the small supercell. This approximation is for ease of calculation due to the challenge of calculating domain boundary movement directly in a large supercell with 960 atoms. Incidentally, we also note that some methods can directly study large-scale systems, such as machine learning algorithms[37,38].

### Translation energy barriers under external electric field

Further, we explore the reasons why the moving barriers of different domain boundaries behave differently under the external electric field. The biggest difference in the movement of the tail-to-tail and head-to-tail domain boundaries is the transition state, where the in-plane electric polarization direction in the tail-to-tail domain boundary is along the *y*-axis, while that in the head-to-tail domain boundary is along the *x*-axis. Based on the above discussion, the in-plane polarization of the two transition states derives from the uniform deviation of the central-layer Se atoms with respect to the symmetry position (the center of the hexagon). Therefore, we focus on the primitive cell with a hexagonal structure. In the primitive cell, the central-layer Se atom is unstable in the symmetry position (the center of the hexagon). It always tends to deviate from a distance of about 0.5 Å to a stable structure, as shown in Supplementary Fig. 8, which is consistent with previous work[39]. Although the energies of these stable structures are almost degenerate and the magnitudes of the in-plane electric polarizations are similar[39], the out-of-plane electric polarizations are significantly different, as shown in Fig. 4d, which can be divided into two categories according to their stable structures with or without out-of-plane electric polarization. One is that when the central-layer Se atoms move to the stable position along the vertical bisector of the sides of the hexagon, the out-of-plane electric polarization of the stable structure is almost zero, which corresponds to the transition state of the tail-to-tail domain boundary (the atomic structure in the blue rectangle of Fig. 4d). Since there is no out-of-plane electric polarization in the initial and final states[29], the energy barrier (the energy difference between the transition state and the initial state) is hardly affected by the external electric field in the *z*-axis. However, the out-of-plane electric polarization of the transition state in the tail-to-tail domain boundary is not strictly zero, about 0.001 eÅ per In$_2$Se$_3$ (1.2 × 10$^{-4}$ C m$^{-2}$), which results in a slight increase in the energy barrier as the external electric field increases (the blue line in Fig. 4c). The other is that when the central-layer Se atoms move to the stable position along the diagonal of the hexagon, the magnitude of the out-of-plane electric polarization in the stable structure is about 0.016 eÅ per In$_2$Se$_3$ (1.9 ×

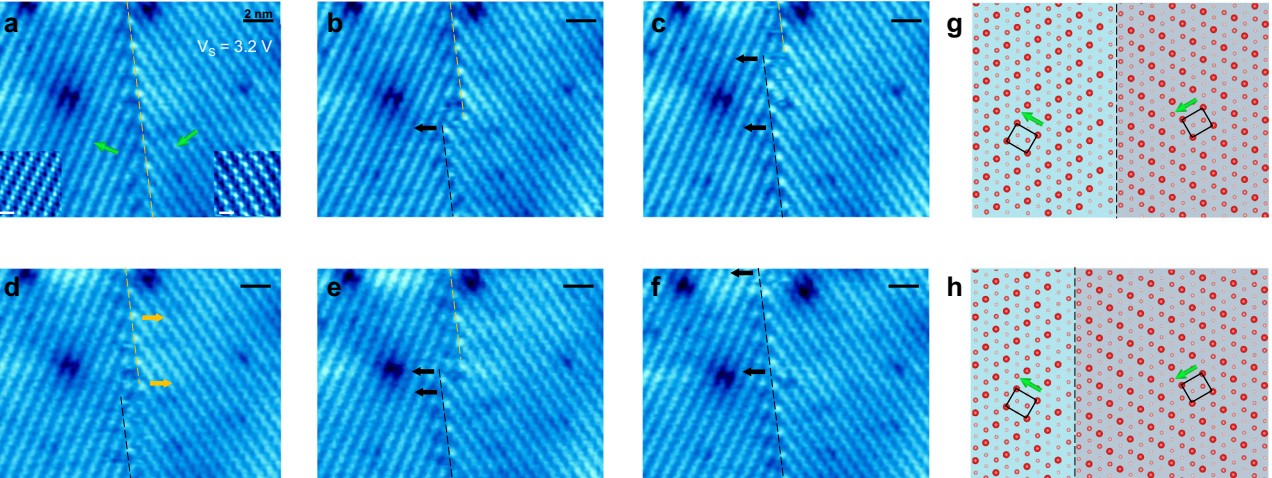

**Fig. 3 | Manipulation of a head-to-tail domain boundary in monolayer $\beta'$ In₂Se₃.**
**a**–**f** STM images showing boundary movements with scanning bias $V_s = 3.2$ V. The initial position of the boundary is indicated by a yellow dashed line, while the final position is indicated by a black dashed line. The black and yellow arrows

respectively represent the left and right movements of part of the boundary. Scanning parameters for **a**–**f**: V = 3.2 V, I = 0.2 nA, 600 s per image. **g**–**h** calculated atomic models of a 60° head-to-tail domain boundary with marked unit cells and polarization.

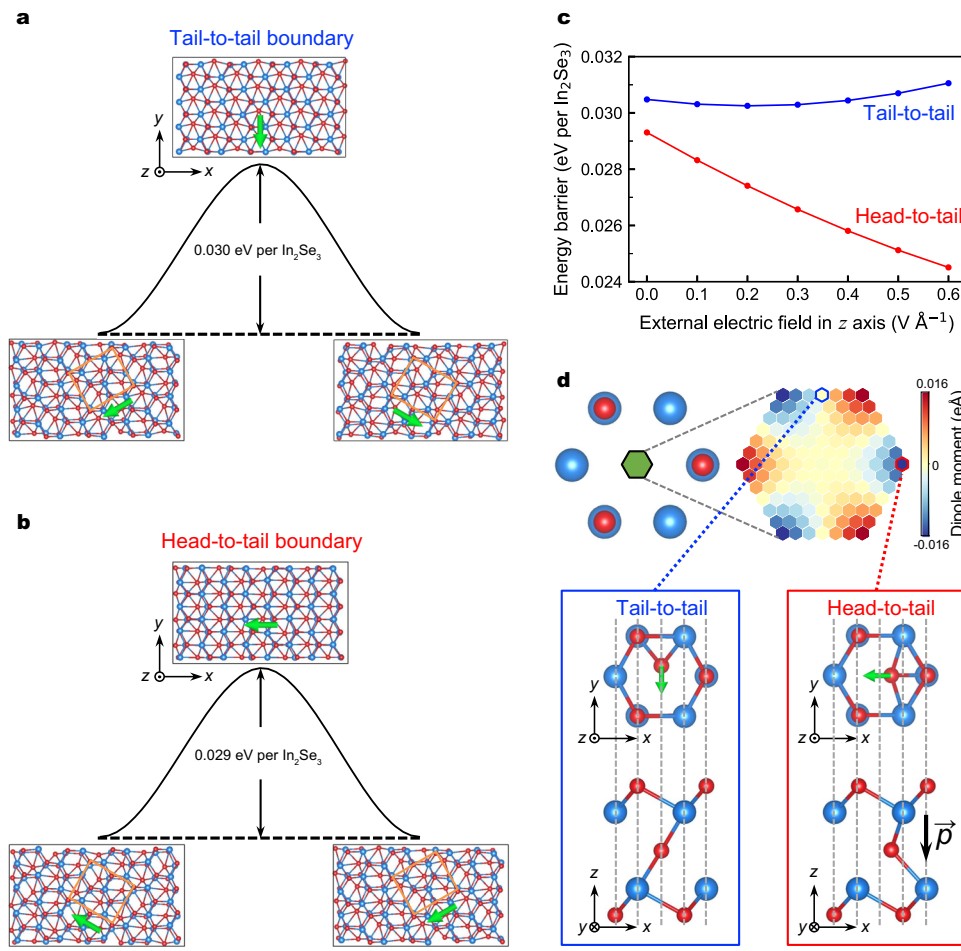

**Fig. 4 | Computational simulation of domain boundary movement.** Energy barrier of simulated **a** tail-to-tail and **b** head-to-tail domain boundary shift. **c** Dependence of the energy barrier on the external electric field, where the energy barrier was calculated by the energy difference between the transition state and the initial state in (**a**, **b**). **d** Map of the out-of-plane dipole moment of the middle Se

atom sampled inside the green hexagon. The corresponding atomic structures of the transition states of the tail-to-tail and head-to-tail domain boundaries are shown in blue and red rectangles, respectively. Green arrows in all atomic structures indicate the directions of the in-plane electric polarization. The black arrow indicates the out-of-plane electric polarization.

$10^{-3}$ C m$^{-2}$), which corresponds to the transition state of the head-to-tail domain boundary (the atomic structure in the red rectangle of Fig. 4d). The existence of the out-of-plane electric polarization makes the transition state of the head-to-tail domain boundary more sensitive to the external electric field, so the energy barrier decreases rapidly with the increase of the electric field.

In summary, we demonstrate the capability to manipulate various types of polar domain boundaries in ferroelectric monolayer In$_2$Se$_3$ by using STM and visualize the evolution of the domain boundaries with atomic precision. We show that the movements of the domain boundaries can be driven by the electric field from an STM tip and proceed by the collective shifting of atoms at the domain boundaries. 60° tail-to-tail domain boundaries have a larger threshold manipulation bias of around 3.4 V compared with the threshold bias of around 2.1 V for head-to-tail domain boundaries. Our DFT calculations reveal the energy barriers of the polarization switch during boundary manipulation and explain the manipulation threshold bias difference between tail-to-tail and head-to-tail domain boundaries. Understanding the domain boundary dynamics and manipulating different types of polar domain boundaries in ferroelectric materials may provide critical insights for future domain boundary based electronic or quantum devices like nanoscale memory[40] or switch[7,9].

## Methods

### In$_2$Se$_3$ synthesis
The In$_2$Se$_3$ nanosheets were synthesized on highly oriented pyrolytic graphite (HOPG) substrates using a chemical vapor deposition method within a homemade tube furnace featuring multiple temperature zones. High-purity selenium powder (Alfa Aesar, 99.999% purity) was evaporated at a stabilized temperature of 270 °C upstream, while In$_2$O$_3$ powder (Alfa Aesar, 99.99% purity) and HOPG substrates (SPI Supplies, USA) were positioned at the upstream in a quartz tube at temperatures of 750 °C and 640 °C, respectively. The vapors of selenium and In$_2$O$_3$ were carried to the HOPG substrate with a gas flow of 20 sccm (Ar:H$_2$ ~ 4:1), regulated by a mass-flow controller.

### STM characterization
STM imaging and manipulation were conducted in an ultra-high vacuum (UHV) STM system (a customized Omicron LT STM/Q-plus AFM system). After transferred into the preparation chamber of the STM system, which maintained a base pressure of < $10^{-10}$ mbar, the samples underwent annealing at 380 °C for 2 h. Subsequently, they were transferred in vacuum to the STM chamber connected to the preparation chamber. An optical microscope (Infinity K2) affixed to one of the optical windows of the STM chamber facilitated the precise positioning of the STM tip onto the desired areas of the samples. STM imaging and manipulation were performed in the STM chamber at 77 K.

### DFT calculation
All the density functional theory calculations were performed using the Vienna Ab Initio Simulation Package (VASP)[41], based on the plane wave basis sets with the projector-augmented wave (PAW) method[42,43]. The exchange and correlation interactions were treated by the Perdew-Burke-Ernzerhof (PBE)[44] parameterization of generalized gradient approximation (GGA). A vacuum layer larger than 15 Å was adopted in our calculations to model the 2D films. The Γ-centered k-point meshes of $20 \times 20 \times 1$ and $2 \times 3 \times 1$ were used for Brillion zone sampling to calculate the map of out-of-plane dipole moment in the primitive cell with a hexagonal structure and the energy barrier of the $\beta'$ In$_2$Se$_3$ supercell, respectively, where the structure of the supercell is shown in the initial or final state in Fig. 4a, b. The cutoff energy of the plane wave basis was set to 250 eV. The convergence criterion of the total energy was $10^{-6}$ eV, and the force convergence criterion was set as 0.01 eV Å$^{-1}$ on each ion. The climbing image nudged elastic band (CI-NEB) method was used to determine the energy barriers of the kinetic processes[45], in

which five intermediate structures were inserted between the initial and final states. All the optimized structures were shown in Supplementary Movies 2, 3, in which each step of the movement involves the collaborative motion of many atoms, and the corresponding energies were shown in Supplementary Fig. 9, where the third intermediate state reaches the saddle point of the most effective kinetic pathway.

### Numerical simulations of electric field
The COMSOL software was used to numerically simulate the tip-induced electric field, where the electric field was set as static. The calculated results are shown in Supplementary Fig. 7, including the parameters of the model.

## Data availability
All data that support the findings of this paper are available within the paper and its Supplementary Information files.

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

## Acknowledgements

F.Z. and C.T. acknowledge the financial support provided for this work by the U.S. Army Research Office under Grant W911NF-15-1-0414. L.L. and A.N. acknowledge support from the National Natural Science Foundation of China (Grant No. 51732010). Z.W. and W.Z. acknowledge the National Key Research and Development Program of China (Grant No. 2019YFA0210004), the Strategic Priority Research Program of Chinese Academy of Sciences (Grant No. XDB30000000), and the Fundamental Research Funds for the Central Universities (Grant No. WK3510000013).

## Author contributions

C.T. conceived and designed the research. F.Z. and C.T. performed the experiments and analyzed the data. Z.W. and W.Z. carried out the theoretical calculations. L.L. synthesized In$_2$Se$_3$ under the supervision from A.N. and Y.G. F.Z. and Y.L. performed the COMSOL simulation. F.Z. and C.T. wrote the paper with support from Z.W. and W.Z. All authors discussed the results and contributed to manuscript revisions.

## Competing interests

The authors declare no competing interests.
