## [Peer Review File · Nature Communications]

Atomic-Scale Manipulation of Polar Domain Boundaries in Monolayer Ferroelectric In₂Se₃REVIEWER COMMENTS

Reviewer #1 (Remarks to the Author):

This work demonstrates controllable manipulation of the domain boundaries in 2D ferroelectric In_2Se_3 by the electric field from an STM tip. The energy path and evolution of the domain boundary movement are revealed with DFT calculations. This topic is important, and the reported results are meaningful to deepen the understanding of domain boundaries in emerging 2D ferroelectric materials. The reviewer recommends accepting this manuscript after addressing the comments below:

(1) The step-by-step movement of domain boundaries based on individual atomic movements is an interesting concept that has also been discussed in a ferroelectric 2D phase of Ga_2O_3 (Phys. Rev. B 2021, 104, 054107, DOI: 10.1103/PhysRevB.104.054107). It is recommended to discuss the similarities and differences between the two systems. Furthermore, the existing ML algorithm of In_2Se_3 (Phys. Rev. B 2021, 104, 174107, DOI: 10.1103/PhysRevB.104.174107) is also worthy of discussion concerning the large-scale simulations.

(2) Figure S1 in the ESI and Figure 1a in the main text display essentially the same configuration, but Figure S1 contains more helpful information. It would be meaningful to merge Figure 1a with Figure S1 in the main text. Additionally, the current Se color coding makes it difficult to determine the orientation between the theoretical configurations and the STM images (in Figures 1a, b, and c).

(3) More information and discussion regarding the NEB calculation should be provided. For instance, including more NBE image points of the intermediate states besides the saddle point would be helpful. Additionally, it is important to note that a transition barrier calculated from a supercell can involve multiple atomic movements either sequentially or collectively. This will result in a highly complex minimum energy pathway with possibly many intermediate minima.

(4) The intrinsic lowering effect of the transition barrier rooted in the GGA-PBE function should be discussed (such as in Science 2008, 321, 792–794, DOI: 10.1126/science.1158722).

(5) The Method section should provide more detailed information regarding the supercell size used in the NBE calculations, as this relates to comment 2.

Reviewer #2 (Remarks to the Author):

The manuscript “Atomic-Scale Manipulation of Polar Domain Boundaries in Monolayer Ferroelectric In_2Se_3 ” by Fan Zhang et al. reports a scanning tunneling microscopy study of the dynamics of ferroelectric domain boundaries in beta- In_2Se_3 . While I find the resolution of a step-by-step movement of domain boundary interesting, the authors need to specify the novelty of this work better, as well as addressing the concerns below, before this manuscript can be published in Nature Communications.

1. Page 3, line 45: To my knowledge, manipulating the ferroelectric domain boundaries by a biased STM tip has been demonstrated in several other works, including K. Chang et al.,

Science 353, 274 (2016); Z. Chen et al., Advanced Science 8, 2100713 (2021); J. Gou et al., Nature (2023) (just published online) <https://doi.org/10.1038/s41586-023-05848-5>. Especially, the work by Z. Chen et al. also studied the domain movements in In_2Se_3 , including not only the switching of polarization in the beta' phase, but also the switching between beta' and beta'' phases. The authors should specify the novelty of their work beyond that of Z. Chen et al.

2. Page 4, line 76: The polarization magnitude " $1.517\text{eA}/\text{In}_2\text{Se}_3$ " is confusing. Do the authors mean "per In_2Se_3 unit cell"? The authors are encouraged to translate it into an exact value in the unit of "C/m", which can be directly compared with other 2D ferroelectrics with in-plane polarization. Similar question also applies to the values in lines 203 and 206 in page 10, except that the unit of " C/m^2 " should be used for out-of-plane polarization.

3. Page 5, line 82: "The majority of the observed domain boundaries are 60° and 120° , with a distribution of $\sim 72\%$ for the 60° domain boundaries and $\sim 27\%$ for the 120° ones." Do the authors mean there are 1% domain boundaries that are neither 60 degree nor 120 degree, since $72\% + 27\% = 99\%$?

4. Page 6, line 103: "parts of the domain boundary usually move first, and then the whole boundary reaches the new position" Consider such a scenario: the domain boundary moves by $2b$ distance towards a certain direction (say left) during the scan, and then moves backward (say right) after several scan lines. This scenario can also explain the "partial domain boundary movement" in Fig. S3. Can the authors distinguish a real partial movement and a back-and-forth movement of a whole domain boundary during one imaging process? For example, if the areas like those in Fig. S3 are scanned with a lower bias voltage that cannot trigger a domain boundary movement, would the authors see repeatable shapes of domain boundaries?

5. Page 6, line 106: "That implies a slight difference between the energy barriers for the transition of the domain boundary to the left and the right." Can the authors characterize such an energy barrier asymmetry?

6. Page 6, line 120: "It is worth noticing that the collective movement of the domain boundary can overcome some point defects" This statement seems contradict the partial domain boundary movement stated in Fig. S3.

Reviewer #3 (Remarks to the Author):

This manuscript investigates the electronic field manipulation of the domain boundaries in 2D ferroelectric In_2Se_3 with by STM. Various types of domain boundaries are observed, with the domain boundaries moving in a step-by-step manner by STM tip. Further DFT calculations explain the different energy paths and evolution movements for the 60° tail to tail and head to tail domain boundaries.

The manuscript provides detailed dynamic behaviors of domain boundaries at atomic scale. However, the main problem is lacking novelty. In fact, the main conclusion of this paper had been reported in their previous study [J. Phys. Chem. Lett. 12, 11902 (2021)], including the atomic structure and the calculated atomic model of domain boundaries, the displacement of

Se atoms in the surface and central layers. The only new findings are the additional 60° head to tail domain boundaries, and its explanation for a lower threshold manipulation bias than the tail to tail case, which is suitable for a more specific journal but cannot satisfy the high standard of Nature Communications. There are a few minor issues that the authors may consider:

(1) Fig. 2 and Fig.3 are bit redundant, since the main features are the same, except for the details of the calculated atomic models.

(2) Previously, the author claimed that the atomic structure transforms nearly to the β phase at the center region of the domain boundary. Since many studies indicate the β - In_2Se_3 is anti-ferroelectric [Phys. Rev. Lett. 125, 047601 (2020), Adv. Mater. 34, 2106951 (2022)]. How this anti-ferroelectricity becomes accommodative with the ferroelectric β' - In_2Se_3 in such a narrow region of domain boundaries? The step length between the nearest neighboring positions is merely $2b$, which is apparently too small to support the anti-ferroelectricity of β' - In_2Se_3 .

Point by Point Response to Reviewers' Comments

Reviewer #1 (Remarks to the Author):

This work demonstrates controllable manipulation of the domain boundaries in 2D ferroelectric In₂Se₃ by the electric field from an STM tip. The energy path and evolution of the domain boundary movement are revealed with DFT calculations. This topic is important, and the reported results are meaningful to deepen the understanding of domain boundaries in emerging 2D ferroelectric materials. The reviewer recommends accepting this manuscript after addressing the comments below:

(1) The step-by-step movement of domain boundaries based on individual atomic movements is an interesting concept that has also been discussed in a ferroelectric 2D phase of Ga₂O₃ (Phys. Rev. B 2021, 104, 054107, DOI: 10.1103/PhysRevB.104.054107). It is recommended to discuss the similarities and differences between the two systems. Furthermore, the existing ML algorithm of In₂Se₃ (Phys. Rev. B 2021, 104, 174107, DOI: 10.1103/PhysRevB.104.174107) is also worthy of discussion concerning the large-scale simulations.

Reply: Thanks to the reviewer for providing these references to give us more inspiration. In the PRB paper about Ga₂O₃ system, employing CI-NEB, CI-(SS)NEB, and the GAP methods they developed, the authors studied in detail the processes of polarization reversal and phase transitions, as well as their barriers at different scales. Among them, the supercells containing different domains were constructed to simulate the movement of domain boundaries. In our work, the supercells with different domain boundaries were also constructed to simulate experimental domain boundaries, but not the movement of domain boundaries (due to the large size of the supercells, containing 960 atoms). Here, by a simple analysis, we find that it is reasonable to approximate the energy barrier of the domain boundaries shift in the large supercell by in-plane polarization inversion in a small supercell with 160 atoms. Therefore, we only calculated the inversion between different polarization directions in a small supercell, similar to the direct process of Fig. 2 in the PRB of Ga₂O₃, except that more atoms are involved in the movement in our work. We believe that our calculations are sufficient to illustrate the differences in the movement of different domain boundaries, but machine learning algorithms are indeed a very advantageous means to directly study the barriers of polarization reversal and phase transitions in large-scale systems, as mentioned in the two PRB papers.

Detailed changes in the main text (highlighted in red) include:

On page 9

This result is consistent with the experimental **phenomenon** that the head-to-tail domain boundary is easier driven by the external electric field, **which also confirms the validity of our previous analysis that approximates the energy barrier of domain boundary shift in the large supercell by in-plane polarization reversal in the small supercell. This approximation is for ease of calculation due to the challenge of calculating domain boundary movement directly in a large supercell with 960 atoms. Incidentally, we also note that some methods can directly study large-scale systems, such as machine learning algorithms^{37,38}.**

(2) Figure S1 in the ESI and Figure 1a in the main text display essentially the same configuration, but Figure S1 contains more helpful information. It would be meaningful to merge Figure 1a with Figure S1 in the main text. Additionally, the current Se color coding makes it difficult to determine the orientation between the theoretical configurations and the STM images (in Figures 1a, b, and c).

Reply: Thanks for the suggestion. We have merged the two figures in the revised manuscript, as shown in Fig. R1 and Fig. 1. In order to see the polarization direction more intuitively, we have changed the color coding of the surface Se layer so that the two Se atoms' height difference becomes apparent inside the unit cell. The surface Se atoms in a unit cell are at different heights, with the highest ones at the corners of a unit cell. Two central Se atoms inside a unit cell arrange in a way that one is higher than the other, and the polarization direction points from the higher atom towards the lower atom.

Fig. R1. Atomic structure of β' In_2Se_3 structure and 60° domain boundary. (a) Atomic model of β' In_2Se_3 structure. Red dashed circles indicate the symmetry position of central layer Se atoms. Black arrows denote dipoles induced by the central layer Se atoms deviating from the symmetry position. The green arrow indicates the directions of the in-plane electric polarization. (b) An atomically resolved STM image of monolayer β' In_2Se_3 ($V_S = 1.0$ V, $I = 0.4$ nA). A unit cell is marked and overlapped with the surface Se atoms from the atomic model, highlighted by the dashed rectangle in the side view in a. The shades of the color indicate the relative height of each Se atom. (c) Illustration of the 60° tail-to-tail domain boundary. The blue dashed lines are along the b directions of the unit cell in each domain, forming a 60° angle. The boundary serving as the angle bisector is marked with black dashed lines. The angle between the polarization vectors of adjacent domains is indicated in green. (d) Large scale STM image showing a 60° domain boundary in monolayer In_2Se_3 ($V_S = 4.2$ V, $I = 0.2$ nA).

(3) More information and discussion regarding the NEB calculation should be provided. For instance, including more NBE image points of the intermediate states besides the saddle point would be helpful. Additionally, it is important to note that a transition barrier calculated from a supercell can involve multiple atomic movements either sequentially or collectively. This will result in a highly complex minimum energy pathway with possibly many intermediate minima.

Reply: Thanks for the constructive comment. For all transition state calculations, we inserted five intermediate states between the initial and final states. Using standard CI-NEB calculations, we investigated the kinetic pathways of polarization reversal to simulate the tail-to-tail and head-to-tail domain boundary shifts. All the optimized structures were shown in Movies 1 and 2 of SI, and the corresponding energies were shown in Fig. R2, where the third intermediate state reaches the saddle point of the most effective kinetic pathway. In addition, the comment mentioned that there may be many intermediate minima for calculating transition barriers involving the movement of multiple atoms in a supercell, since the atoms may move sequentially or collectively. However, in our calculated systems, the displacement of the individual atoms between the initial and final states was small (most atoms move less than 1 Å), perturbing only in a small region, as shown by the atoms in the green circle in the Movies 1 and 2 of SI. Meanwhile, each step of the movement involves the collaborative motion of many atoms, and thus there is only one extreme value in our results.

Fig. R2. Energy barrier of simulated tail-to-tail and head-to-tail domain boundary shift.

Detailed changes in the main text (highlighted in red) include:

On page 14

The climbing image nudged elastic band (CI-NEB) method was used to determine the energy barriers of the kinetic processes⁴⁵, in which five intermediate structures were inserted between the initial and final states. All the optimized structures were shown in Movies 1 and 2 of SI, in which each step of the movement involves the collaborative motion of many atoms, and the corresponding energies were shown in Figure S9, where the third intermediate state reaches the saddle point of the most effective kinetic pathway.

(4) The intrinsic lowering effect of the transition barrier rooted in the GGA-PBE function should be discussed (such as in Science 2008, 321, 792–794, DOI: 10.1126/science.1158722).

Reply: Thanks for the comment. To answer this question, we carefully reviewed relevant literature, including the Science paper mentioned here. To the best of our knowledge, the underestimation of barriers in DFT discussed in most papers is limited to chemical reactions, due to the electrons of transition state delocalized over more than one center. However, we are unaware of any relevant discussions for structural

phase transitions or polarization reversal due to atomic displacements. On the other hand, we are interested in the response of the energy barrier of the different domain boundary movement to the external electric field. Specifically, our calculations show that the energy barrier of the domain boundary movement at the tail-to-tail is insensitive to the external electric field, while it decreases rapidly as the electric field increases at the head-to-tail. Even if the DFT does underestimate the energy barrier, it does not affect our qualitative conclusions, except for a possible quantitative change.

(5) *The Method section should provide more detailed information regarding the supercell size used in the NBE calculations, as this relates to comment 2.*

Reply: Thanks to the reviewer for the reminder. In order to simulate the experimental structure of 60° tail-to-tail domain boundary, we constructed a supercell with 960 atoms, as shown in Fig. R3, which has been detailed described in our previously published work (J. Phys. Chem. Lett. 2021, 12, 11902). As we mentioned in the Reply of Comment 1, due to the large size of the supercell, it is challenging to calculate the domain boundary movement directly in this structure [Fig. R3(b)]. By a simple analysis, we find that it is reasonable to approximate the energy barrier of the domain boundaries shift in the large supercell [Fig. R3(b)] by in-plane polarization inversion in a small supercell with 160 atoms [Orange rectangle in Fig. R3(a)].

Fig. R3. Supercell structures of a 60° tail-to-tail domain boundary. (a) Unit cells of β' - In_2Se_3 (small black rectangles) with polarization along the horizontal direction. A built unit (orange rectangle) to simulate angled domain boundaries. (b) Schematic of the geometry of the domain boundary structure with 60° as defined in the experiment. The left domain and the right domain are respectively from the left built unit and the right built unit in (a). Green arrows denote the polarization directions in all figures.

Detailed changes in the main text (highlighted in red) include:

On page 13-14

A Γ -centered $20 \times 20 \times 1$ and $2 \times 3 \times 1$ k-point meshes were used for Brillion zone sampling to calculate the map of out-of-plane dipole moment in the primitive cell and the energy barrier of the supercell, respectively, where the structure of the supercell is referred to our previous work³⁶.

Reviewer #2 (Remarks to the Author):

The manuscript “Atomic-Scale Manipulation of Polar Domain Boundaries in Monolayer Ferroelectric In₂Se₃” by Fan Zhang et al. reports a scanning tunneling microscopy study of the dynamics of ferroelectric domain boundaries in beta-In₂Se₃. While I find the resolution of a step-by-step movement of domain boundary interesting, the authors need to specify the novelty of this work better, as well as addressing the concerns below, before this manuscript can be published in Nature Communications.

I. Page 3, line 45: To my knowledge, manipulating the ferroelectric domain boundaries by a biased STM tip has been demonstrated in several other works, including K. Chang et al., Science 353, 274 (2016); Z. Chen et al., Advanced Science 8, 2100713 (2021); J. Gou et al., Nature (2023) (just published online) <https://doi.org/10.1038/s41586-023-05848-5>. Especially, the work by Z. Chen et al. also studied the domain movements in In₂Se₃, including not only the switching of polarization in the beta' phase, but also the switching between beta' and beta'' phases. The authors should specify the novelty of their work beyond that of Z. Chen et al.

Reply: We appreciate the reviewer for raising this issue and mentioning related works. Demonstration and understanding the mechanism for manipulating polar domain boundaries in 2D ferroelectric materials have always been critical, especially for domain boundary based nanoelectronic device applications. As the reviewer mentioned, those works demonstrate domain switching or changing polarization in ferroelectric materials with the electric field induced by an STM or AFM tip. However, we are focusing on the detailed domain boundary movement or dynamics at the atomic scale, which is missing in previous reports. We systemically investigate the STM tip bias difference (electric field) for different types of domain boundaries with detailed statistics (we add Figure S6 to SI). The calculated electric field dependent energy barrier difference for manipulating the tail-to-tail (head-to-head) and head-to-tail boundary further supports our observation. We made some comparisons of those works with ours below and added more details in the revised manuscript.

K. Chang et al. reported the discovery of in-plane ferroelectricity in atomic thick SnTe and manipulation of the domains after a high-voltage pulse. The domain switching supported the ferroelectricity's existence. J. Gou et al. demonstrated the in-plane ferroelectricity in black phosphorous-like Bi monolayer. They did the sample bias sweeping (negative to positive or positive to negative) to switch the in-plane polarizations and thus move the domains. It's a well-controlled domain switching in 2D ferroelectrics.

Regarding the studied material, indeed, Z. Chen's work is closest to ours, both on 2D In₂Se₃. The main results in their Advanced Science paper are (i) the reversible switching of the β' phase from one direction to another, and (ii) the reversible switching between the β' (we call it β) and β'' (we call it β') domains using different bias scanning (same method as us). Their demonstration shows the feasibility and advantages of manipulating domain boundary movement with the electric field from the STM tip during scanning. In fact, before their work, our previous works have revealed the thermally driven phase transition between β' and β'' phase (ACS Nano 13, 7, 8004 (2019)) and demonstrated scanning induced β' and β'' domain movements (Fig. S4 in JPCL 12, 11902 (2021)). Based on our previous work, our current work focuses on the manipulation of different types of β'' domain boundaries (tail-to-tail, head-to-head, and head-

to-tail), dynamics of the boundary movement and underlying mechanism of different switching behaviors with corroborative theoretical calculations.

In summary, compared with previous works focusing on the initial and final state of the boundary, we are focusing on the polar domain boundaries and their movement or detailed domain boundaries dynamics. This domain boundary dynamic process will deepen our understanding of the interplay between material atomic structure and polarization switch induced by a local electric field. We made a movie in SI (SI movie 3) showing the whole process of a boundary movement which exhibits the dynamic process. Our work based on large scale simulation could provide a theoretical way to investigate domain boundaries' structure. Also, we demonstrated the STM tip bias difference (electric field) and calculated the energy barrier difference for manipulating the tail-to-tail and head-to-tail boundary. Distinguishing the different switching behavior for these two types of domain boundaries will be helpful for future domain boundary based nanoelectronics.

Detailed changes in the main text (highlighted in red) include:

On page 3

Several recent studies demonstrated the possibility of tip bias induced domain or polarization switching in 2D ferroelectric materials^{22,25-27}. Here we used STM to manipulate the domain boundaries in 2D In₂Se₃, an emerging 2D ferroelectric material²⁸⁻³⁵. Atomically thin In₂Se₃ was synthesized through the chemical vapor deposition (CVD) method. We manipulate various types of domain boundaries in monolayer β' In₂Se₃ with atomic precision and study the detailed domain boundary dynamics during manipulation by using STM.

On page 12

Understanding the domain boundary dynamics and manipulating different types of polar domain boundaries in ferroelectric materials may provide critical insights for future domain boundary based electronic or quantum devices like nanoscale memory⁴⁰ or switch^{7,9}.

2. Page 4, line 76: The polarization magnitude “1.517eA/In2Se3” is confusing. Do the authors mean “per In2Se3 unit cell”? The authors are encouraged to translate it into an exact value in the unit of “C/m”, which can be directly compared with other 2D ferroelectrics with in-plane polarization. Similar question also applies to the values in lines 203 and 206 in page 10, except that the unit of “C/m^2” should be used for out-of-plane polarization.

Reply: We thank the reviewer for this suggestion. The unit “dipole moment per In₂Se₃” means “per In₂Se₃ unit cell”. We used that unit in the original manuscript because it is convenient and unambiguous for the calculation and consistent with our previous work (ACS Nano 13, 8004 (2019); J. Phys. Chem. Lett. 12, 11902 (2021)). There is some ambiguity/arbitrariness in the definition of polarization in 2D materials. For example, both units “C/m” and “C/m²” are used for the magnitude of in-plane (out-of-plane) polarization (Adv. Electron. Mater. 6, 1900818 (2020)). Following the reviewer’s suggestion for consistency and accuracy, we now adopt a commonly used unit of polarization, which is the number of dipole moments per

unit area (C/m). In the revised manuscript, we have added the data in this unit after each electric dipole moment.

Detailed changes in the main text (highlighted in red) include:

On page 5

$1.517 \text{ e}\text{\AA}/\text{In}_2\text{Se}_3$ ($1.7 \times 10^{-10} \text{ C/m}$)

On page 11

$0.001 \text{ e}\text{\AA}/\text{In}_2\text{Se}_3$ ($1.1 \times 10^{-13} \text{ C/m}$)

$0.016 \text{ e}\text{\AA}/\text{In}_2\text{Se}_3$ ($1.8 \times 10^{-12} \text{ C/m}$)

3. Page 5, line 82: “The majority of the observed domain boundaries are 60° and 120° , with a distribution of $\sim 72\%$ for the 60° domain boundaries and $\sim 27\%$ for the 120° ones.” Do the authors mean there are 1% domain boundaries that are neither 60 degree nor 120 degree, since $72\% + 27\% = 99\%$?

Reply: Thanks for pointing this out. The rest 1% domain boundaries are 180° domain boundaries. Shown in Fig. R4 is an STM image of the domain boundary. We also added it as Fig. S1 in the revised SI.

Fig. R4. An STM image of an 180° head-to-tail domain boundary ($V = 1.5 \text{ V}$, $I = 0.4 \text{ nA}$).

Detailed changes in the main text (highlighted in red) include:

On page 5

Rarely, 180° domain boundaries (shown in Figure S1) were observed, which account for $\sim 1\%$ of all the observed domain boundaries.

4. Page 6, line 103: “parts of the domain boundary usually move first, and then the whole boundary reaches the new position” Consider such a scenario: the domain boundary moves by $2b$ distance towards a certain direction (say left) during the scan, and then moves backward (say right) after several scan lines. This scenario can also explain the “partial domain boundary movement” in Fig. S3. Can the authors distinguish a real partial movement and a back-and-forth movement of a whole domain boundary during one imaging process? For example, if the areas like those in Fig. S3 are scanned with a lower bias voltage that cannot trigger a domain boundary movement, would the authors see repeatable shapes of domain boundaries?

Reply: We appreciate the reviewer’s question on the details of domain boundary moving dynamics. Although we didn’t do exactly at the boundary shown in Fig. S3, we did a careful bias dependent manipulation on various domain boundaries, confirming our claim. Fig. R5 (also Figure S5) shows a sequence of STM images for a head-to-head domain boundary under different manipulation conditions (scanning bias). As shown in Fig. R5a-d, the domain boundary movement was not triggered when the bias was lower than (or equal) 3.5 V. When the bias increased to 4.5 V, the boundary started moving, as shown in Fig. R5e-h. When we decreased the bias to 3.5 V, the boundary stopped moving, as shown in Fig. R5i-l. In summary, the repeatable shape of the domain boundary is observed when the bias is lower than the threshold bias that triggers a domain boundary movement. We also did a statistic of the threshold bias for several different types of domain boundaries, as shown in Fig. R6.

Fig. R5 (Same as Fig. S5) Domain boundary dynamics for a head-to-head domain boundary. (a-l) a sequence of STM images for the same head-to-head domain boundary. The scanning bias for each image is labeled above.

Fig. R6 (same as Fig. S6). The statistic of the threshold bias for manipulating different types of domain boundaries. The statistic is derived from the manipulation results of two different tail-to-tail domain boundaries, four different head-to-head domain boundaries, and four head-to-tail domain boundaries.

Detailed changes in the main text (highlighted in red) include:

On page 8

With similar domain boundary moving dynamics, the threshold bias for 60° head-to-head domain boundary (Figure S5) is close to that for tail-to-tail type. In our experiments, we observed that the averaged threshold bias for the 60° head-to-tail domain boundaries is around 2.1 V, while for the 60° tail-to-tail (or head-to-head) domain boundaries, the averaged threshold bias is around 3.4 V (see more detailed statistic in Figure S6).

5. Page 6, line 106: “That implies a slight difference between the energy barriers for the transition of the domain boundary to the left and the right.” Can the authors characterize such an energy barrier asymmetry?

Reply: Thanks for the comment. As shown in Fig. R7, both the theoretical calculations and STM imaging show that the structures on both sides of the domain boundary are not mirror-symmetric (see details in our previous work: J. Phys. Chem. Lett. 12, 11902 (2021)). The difference between the energy barriers for the transition of the domain boundary to the left and the right may also be due to this structural asymmetry.

Also, this structural asymmetry can be described with the displacement of relaxed Se atoms from referenced atoms in the central Se layer, as shown in Fig. R8 below. The displacement is asymmetric at two sides of

the boundary, resulting in asymmetry in the local polarization close to the boundary. This asymmetric local polarization could be the main cause for the energy barrier asymmetry.

Fig. R7 (a) Calculated atomic model of a 60° tail-to-tail boundary. The red atoms represent the Se atoms (b) STM images of a 60° tail-to-tail boundary overlapped with the atomic model. (same as Fig. 2g).

Fig. R8 Displacement of Se atoms in the central layers. Blue atoms represent the relaxed Se atoms, and purple ones represent the reference atoms. The bottom curves represent the displacements of Se atoms in the x, y, and z directions, respectively. The green arrows indicate the direction and magnitude of the polarization as a function of the distance from the domain boundaries. The red arrows indicate the change of polarization of the unit cells in the relaxed atomic structure relative to the reference structure.

6. Page 6, line 120: “It is worth noticing that the collective movement of the domain boundary can overcome some point defects” This statement seems contradict the partial domain boundary movement stated in Fig. S3.

Reply: We thank the reviewer for pointing this out. The original phrasing is indeed a little bit confusing. We are trying to convey that some individual point defects (one highlighted by red circles in Fig. R9 (Fig. 2a-c in the manuscript)) remain at the same location after the boundary moves over, and these defects do not pin the domain boundary. We have corrected the phrasing in the manuscript.

Fig. R9 (Fig. 2a-c in the manuscript) Domain boundary movement of a tail-to-tail domain boundary. The defect in the path is highlighted by solid red circles.

**Detailed changes in the main text (highlighted in red) include:
On page 7**

It is worth noticing that the movement of the domain boundary is not pinned by some types of point defects in the path, one of which is highlighted by the solid red circles in Figures 2a-c.

Reviewer #3 (Remarks to the Author):

This manuscript investigates the electronic field manipulation of the domain boundaries in 2D ferroelectric In₂Se₃ with by STM. Various types of domain boundaries are observed, with the domain boundaries moving in a step-by-step manner by STM tip. Further DFT calculations explain the different energy paths and evolution movements for the 60° tail to tail and head to tail domain boundaries. The manuscript provides detailed dynamic behaviors of domain boundaries at atomic scale. However, the main problem is lacking novelty. In fact, the main conclusion of this paper had been reported in their previous study [J. Phys. Chem. Lett. 12, 11902 (2021)], including the atomic structure and the calculated atomic model of domain boundaries, the displacement of Se atoms in the surface and central layers. The only new findings are the additional 60° head to tail domain boundaries, and its explanation for a lower threshold manipulation bias than the tail to tail case, which is suitable for a more specific journal but cannot satisfy the high standard of Nature Communications. There are a few minor issues that the authors may consider:

Reply: We appreciate the reviewer for raising the concern of the novelty of this work. Demonstration and understanding the mechanism for the manipulation of polar domain boundaries in 2D ferroelectric materials have always been critical, especially for domain boundary based nanoelectronic device applications. Some recent important papers as reviewer 2 mentioned: K. Chang et al., Science 353, 274 (2016); Z. Chen et al., Advanced Science 8, 2100713 (2021); J. Gou et al., Nature 617, 67 (2023). As the reviewer said, our work provides detailed dynamic behavior of domain boundaries at the atomic scale. It demonstrates the manipulation of domain boundaries in ferroelectric materials with the electric field induced by the STM tip. Here we compare their works with ours and explain the dramatic difference with our previous JPCL paper.

K. Chang et al. reported the discovery of in-plane ferroelectricity in atomic thick SnTe and manipulation of the domains after a high-voltage pulse. The domain switching supported the ferroelectricity's existence. J. Gou et al. demonstrated the in-plane ferroelectricity in black phosphorous-like Bi monolayer. They did the sample bias sweeping (negative to positive or positive to negative) to switch the in-plane polarizations and thus move the domains. It's a well-controlled domain switching in 2D ferroelectrics.

Regarding the studied material, Z. Chen's work is closest to ours, both on 2D In₂Se₃. The main results in their Advanced Science paper are (i) the reversible switching of the β' phase from one direction to another, and (ii) the reversible switching between the β' (we call it β) and β'' (we call it β') domains using different bias scanning (same method as us). Their demonstration shows the feasibility and advantages of manipulating domain boundary movement with the electric field from the STM tip during scanning. In fact, before their work, our previous works have revealed the thermally driven phase transition between β' and β'' phase (ACS Nano 13, 7, 8004 (2019)) and demonstrated scanning induced β' and β'' domain movements. (Fig. S4 in JPCL 12, 11902 (2021)). Based on our previous work, our current work focuses on the manipulation of different types of β'' domain boundaries (tail-to-tail, head-to-head, and head-to-tail), dynamics of the boundary movement and underlying mechanism of different switching behaviors with corroborative theoretical calculations.

Our previous JPCL paper merely focuses on the atomic and electronic boundary structure of different types of domain boundary in 2D β' In₂Se₃. It is the necessary preparatory work to the study of the manipulation and domain moving dynamic, which has long been vital to ferroelectric domain boundary based

nanoelectronic applications and is the focus of this manuscript. Compared with previous works focusing on the initial and final state of the boundary, we are focusing on the step-by-step movement or detailed domain boundary dynamics (movement) during the manipulation process. This domain boundary dynamic process will deepen our understanding of interplay between material atomic structure and polarization switch induced by local electric field. We made a movie in SI (SI movie 3) showing the whole process of a boundary movement which clearly exhibits the dynamic process. Our work based on large scale simulation could provide a theoretical way investigating domain boundaries' structure. Also, we demonstrate the STM tip bias difference (electric field) and calculated the energy barrier difference for manipulating the tail-to-tail and head-to-tail boundary. Distinguishing the different switching behavior for these two types of domain boundaries may be helpful for future domain boundary based nanoelectronics.

Detailed changes in the main text (highlighted in red) include:

On page 3

Several recent studies demonstrated the possibility of tip bias induced domain or polarization switching in 2D ferroelectric materials^{22,25-27}. Here we used STM to manipulate the domain boundaries in 2D In₂Se₃, an emerging 2D ferroelectric material²⁸⁻³⁵. Atomically thin In₂Se₃ was synthesized through the chemical vapor deposition (CVD) method. We manipulate various types of domain boundaries in monolayer β' In₂Se₃ with atomic precision and study the detailed domain boundary dynamics during manipulation by using STM.

On page 12

Understanding the domain boundary dynamics and manipulating different types of polar domain boundaries in ferroelectric materials may provide critical insights for future domain boundary based electronic or quantum devices like nanoscale memory⁴⁰ or switch^{7,9}.

(I) Fig. 2 and Fig.3 are bit redundant, since the main features are the same, except for the details of the calculated atomic models.

Reply: We thank the referee for bringing up this issue. From the features of the STM image side, it looks a bit redundant. However, Fig. 2 and Fig. 3 are arranged to show different types of domain boundaries and convey different ideas. Figure 2 shows that the boundary can move all the way to the left under tip manipulation, which agrees excellently with atomic models. Figure 3 also represents the boundary movement but focuses on the detailed dynamic process of the movement. With similar domain boundary moving dynamics to tail-to-tail type, we added a sequence of STM images showing the boundary moving process of a head-to-head boundary in Fig. S5 (shown as Fig. R10 below). Moreover, a detailed statistic of the threshold biases for manipulating different types of boundaries are added in Fig. S6 (shown as Fig. R11 below).

Fig. R10 (same as Fig. S5). Domain boundary dynamics for a head-to-head domain boundary. (a-l) a sequence of STM images for the same head-to-head domain boundary. The scanning bias for each image is labeled above.

Figure R11 (same as Fig. S6). The statistic of the threshold bias for manipulating different types of domain boundaries. The statistic is derived from the manipulation results of two different tail-to-tail domain boundaries, four different head-to-head domain boundaries, and four head-to-tail domain boundaries.

**Detailed changes in the main text (highlighted in red) include:
On page 8**

With similar domain boundary moving dynamics, the threshold bias for 60° head-to-head domain boundaries (Figure S5) is close to that for tail-to-tail type. In our experiments, we observed that the averaged threshold bias for the 60° head-to-tail domain boundaries is around 2.1 V, while for the 60° tail-to-tail (or head-to-head) domain boundaries, the averaged threshold bias is around 3.4 V (see more detailed statistic in Figure S6).

(2) Previously, the author claimed that the atomic structure transforms nearly to the β phase at the center region of the domain boundary. Since many studies indicate the β -In₂Se₃ is anti-ferroelectric [Phys. Rev. Lett. 125, 047601 (2020), Adv. Mater. 34, 2106951 (2022)]. How this anti-ferroelectricity becomes accommodative with the ferroelectric β' -In₂Se₃ in such a narrow region of domain boundaries? The step length between the nearest neighboring positions is merely $2b$, which is apparently too small to support the anti-ferroelectricity of β' -In₂Se₃.

Reply: We thank the reviewer for the constructive comment. In our previous work (ACS Nano 13, 7, 8004 (2019)), we systematically investigated β -In₂Se₃. In our study and those papers that the reviewer mentioned, the anti-ferroelectric β phase always appears as strips with a width of 4 or more unit cells, as shown in Fig. R12 below (image from our paper: ACS Nano 13, 7, 8004 (2019)). The ferroelectricity is within the strip, and the anti-ferroelectricity is between the neighboring strips. We can speculate that if the area of the flake in the direction perpendicular to the strips is reduced to the width of only one nanostrip, then only ferroelectricity may be presented in this narrow strip. In our work, the center region of the domain boundary is only one or two unit cells, which is smaller than the width of an experimental nanostrip, and therefore unlikely exhibits anti-ferroelectricity.

On the other hand, the polarization direction of the domain boundary region is constrained by the polarization direction of the domains on both sides of the domain boundary. For example, Figure R13a shows the atomic model for a tail-to-tail domain boundary. The atomic structure in the central region of the domain boundary is nearly β -phase. As shown in Fig. R13b, the polarization almost vanishes in the x direction (perpendicular to the boundary) while persisting along the y direction (along the boundary). In this case, we think the polarization of the domain boundary center region only exhibits polarization in one direction, like the polarization for a single strip in β -phase.

Fig. R12 A typical atomically resolved STM image showing the strip structure of anti-ferroelectric β - In_2Se_3 . The neighboring two strips show different polarization directions, as marked by the blue arrows ($V_s = 1 \text{ V}$, $I = 0.6 \text{ nA}$).

Fig. R13 (a) Calculated atomic model of a 60° tail-to-tail boundary. (b) Displacement of Se atoms in the central layers. Blue atoms represent the relaxed Se atoms, and purple ones represent the reference atoms. The bottom curves represent the displacements of Se atoms in the x, y, and z directions, respectively. The green arrows indicate the direction and magnitude of the polarization as a function of the distance from the domain boundaries. The red arrows indicate the change of polarization of the unit cells in the relaxed atomic structure relative to the reference structure.

REVIEWER COMMENTS

Reviewer #1 (Remarks to the Author):

The authors have satisfactorily addressed all the concerns raised, and as a result, I recommend that the manuscript be accepted.

Reviewer #2 (Remarks to the Author):

The authors have addressed most of my concerns except the two below. The manuscript can be published after the remaining issues are solved.

Page 11, lines 216 and 219: It is good to see the authors have added the exact values of polarization, but I think the out-of-plane polarization should be specified in the unit of C/m^2 . For in-plane polarization, P can be defined from the 1D bound charge density at the edge of a 2D system, hence in the unit of C/m ; while for out-of-plane polarization P can be defined from the 2D bound charge density at the surface of a 2D system, hence in the unit of C/m^2 . Some of the previous literatures used C/m^2 for in-plane polarization because they simply followed the traditional definition in 3D ferroelectrics, and when they calculate the value of P , they might have obtained the “area of the edge” by multiplying the ill-defined thickness of a 2D film. This is not a strict way of defining in-plane polarization. Nevertheless, for out-of-plane polarization, the unit should always be C/m^2 , no matter for 2D or 3D ferroelectrics.

In response to the previous question #3, the authors have added a new Fig. S1. However, I don't understand what is a “head-to-tail 180° domain boundary”. When people talk about 180° domains in ferroelectrics, they mean the polarization across the domain wall are anti-parallel to each other, and this is what “ 180° ” refers to. In the case of Fig. S1, the polarization direction is not changed across the line structure, so I do not regard it as a “domain boundary”, but rather a structural defect in one whole ferroelectric domain.

Reply to Reviewers' Comments

Reviewer #1 (Remarks to the Author):

The authors have satisfactorily addressed all the concerns raised, and as a result, I recommend that the manuscript be accepted.

[Reply]:

We are grateful to the reviewer and thank him/her for the supportive assessment of our work.

Reviewer #2 (Remarks to the Author):

The authors have addressed most of my concerns except the two below. The manuscript can be published after the remaining issues are solved.

(1) Page 11, lines 216 and 219: It is good to see the authors have added the exact values of polarization, but I think the out-of-plane polarization should be specified in the unit of C/m^2 . For in-plane polarization, P can be defined from the 1D bound charge density at the edge of a 2D system, hence in the unit of C/m ; while for out-of-plane polarization P can be defined from the 2D bound charge density at the surface of a 2D system, hence in the unit of C/m^2 . Some of the previous literatures used C/m^2 for in-plane polarization because they simply followed the traditional definition in 3D ferroelectrics, and when they calculate the value of P , they might have obtained the “area of the edge” by multiplying the ill-defined thickness of a 2D film. This is not a strict way of defining in-plane polarization. Nevertheless, for out-of-plane polarization, the unit should always be C/m^2 , no matter for 2D or 3D ferroelectrics.

[Reply]:

We are grateful to the reviewer for his/her constructive and valuable comments, which have significantly improved our manuscript. We are pleased to note that we have addressed most of the concerns. Here we appreciate the reviewer's patience and thorough explanation regarding the polarization units. In the revised manuscript, we have provided the out-of-plane polarization values in units of C/m^2 .

On page 11

0.001 eÅ/In₂Se₃ ($1.2 \times 10^{-4} C/m^2$)

0.016 eÅ/In₂Se₃ ($1.9 \times 10^{-3} C/m^2$)

(2) In response to the previous question #3, the authors have added a new Fig. S1. However, I don't understand what is a “head-to-tail 180° domain boundary”. When people talk about 180° domains in ferroelectrics, they mean the polarization across the domain wall are anti-parallel to each other, and this is what “180°” refers to. In the case of Fig. S1, the polarization direction is

not changed across the line structure, so I do not regard it as a “domain boundary”, but rather a structural defect in one whole ferroelectric domain.

[Reply]:

We agree that 180° boundaries normally refer to antiparallel polarizations (head-to-head or tail-to-tail), and the description for the new Supplementary Fig. 1 was not sufficiently clear.

As described in detail in our prior publication (ACS Nano 2019, 13, 8004), the β' In_2Se_3 is derived from the β In_2Se_3 through structural phase transformation, where two or more domains of β' In_2Se_3 can come from one domain of β In_2Se_3 . Based on the three-fold symmetry of the β phase, theoretically, 60°, 120°, and 180° domain boundaries are allowed, as shown in the red panels in Fig. R1 below. Considering polarizations, the states in these blue panels in Fig. R1 are not 180° domain boundaries. However, in our experiments, 180° domain boundaries with antiparallel polarizations (head to head or tail to tail) were not observed. The reason may be complex as the formation of domain boundaries can be affected in various ways. For example, as illustrated in Figure R1, the probability of the 180° domain boundaries is lower than that of the 60° or 120° domain boundaries. In addition, the formation energy may vary for each type of domain boundaries. Supplementary Fig. 1 shows one of the observed boundary-like structures beyond the 60° and 120° domain boundaries. It is similar to a 180° domain boundary, but a closer analysis revealed that the polarization directions of the domains on both sides are the same, indicating it is not an actual 180° domain boundary. We concur with the reviewer that the structure can be considered a structural defect. Nevertheless, as shown in Figure R2, the region contains the end part of the boundary in Fig. S1, which separates Domains A and B on each side. So we argue it can also be regarded as a domain boundary, albeit not a 180° domain boundary.

We have revised the caption of Supplementary Fig. 1 and changed the corresponding description in the revised manuscript.

Figure R1. Illustrations of various types of domain boundaries in β' In_2Se_3 . The top panel of each subfigure shows one possibility for the formation of 60°, 120°, and 180° domain boundaries in β' In_2Se_3 after the phase transformation from β to β' phase. The three panels at the bottom of each subfigure show the different polarization configurations in the corresponding 60°, 120°, or 180° domain boundaries, with polar domain boundaries present in the red panels and absent in the blue panels. In all subfigures, the black rectangle denotes the unitcell, the green arrow represents the polarization direction, and the black dashed line indicates the polar domain boundary.

Figure R2. An STM image of a region that consists of three domains, labeled as Domains A, B, and C. The domain boundary between Domains A and B is the end part of the one shown in Supplementary Fig. 1. The red rectangle indicates the unitcell in each domain.

On page 5 of the main text

Rarely, other types of domain boundaries that can not be well defined or can be interpreted as structural line defects that separate domains (one of them shown in Supplementary Fig. 1) were observed, which account for ~ 1% of all the observed domain boundaries.

In the Supporting Information

In the caption of Supplementary Fig. 1, “An STM image of an 180° head-to-tail domain boundary” has been changed to “An STM image of a less clearly defined domain boundary”.

REVIEWERS' COMMENTS

Reviewer #2 (Remarks to the Author):

The authors have addressed all my concerns and the manuscript can be published as it is.